# The Effect of Maternal SARS-CoV-2 Infection on Neonatal Outcome

**DOI:** 10.3390/children10050771

**Published:** 2023-04-25

**Authors:** Melinda Matyas, Madalina Valeanu, Monica Hasmasanu, Bianca Voina, Adelina Tutu, Gabriela C. Zaharie

**Affiliations:** 1Neonatology Department, Iuliu Hatieganu University of Medicine and Pharmacy, 400347 Cluj Napoca, Romania; 2Medical Informatics and Biostatistics Department, Iuliu Hatieganu University of Medicine and Pharmacy, 400347 Cluj Napoca, Romania; 3Neonatology Department, County Emergency Hospital, 400006 Cluj Napoca, Romania

**Keywords:** newborn, COVID-19 infection, breastfeeding

## Abstract

(1) Background: Neonates born to SARS-CoV-2 positive mothers are at risk of infection, as well as adverse outcomes due to the infection. The aim of our study was to analyze the impact of maternal SARS-CoV-2 infection on neonatal outcome. (2) Methods: We conducted a prospective, longitudinal study. We collected data on maternal symptomatology upon admission and their correlation with the development of the infant. Through a questionnaire we analyzed the impact on breastfeeding of the separation of the mother from the newborn, as well as the maternal psycho-emotional effect. (3) Results: Ninety infants were enrolled in the study, from one twin pregnancy and the rest singleton pregnancies. Out of the 89 mothers, 34 showed symptoms. Neonates from mothers with anosmia and ageusia had a higher value of WBC and lymphocytes (*p* = 0.06 and *p* = 0.04). Breastfeeding was started in 57.3% of mothers after their discharge from hospital and only 41.6% of the whole study group continued at the follow-up visit. Mothers who described a negative experience during hospitalization associated a 2.42 times higher risk of not continuing breastfeeding. (4) Conclusion: None of the infants enrolled in the study had SARS-CoV-2 infection either at birth or within the first two months of life. Breastfeeding was started with more than half newborns after discharge from hospital. The negative experience generated by the separation from their babies influenced breastfeeding.

## 1. Introduction 

Due to the immunological peculiarities associated with pregnancy, pregnant women represent a category with increased risk for SARS-CoV-2 infection [1,2,3]. Neonates born to SARS-CoV-2 positive mothers are at risk of infection, as well as adverse outcomes due to the infection [1,2]. The care needs of healthy newborns, but especially of those who are sick and premature and hospitalized in neonatal intensive care units have not been considered adequately during the COVID-19 pandemic, which has created exceptional challenges and disrupted healthcare provision across the globe even more [3,4,5,6]. Several measures were put in place to reduce social interaction and the risk of virus transmission, especially in hospital settings, including maternity and newborn care units [7,8]. In neonatal intensive care units (NICUs), measures aimed at stemming transmission were implemented, which had immediate adverse consequences on the care of the most vulnerable groups of patients—sick, preterm, and low birthweight infants.

Recent research demonstrates that the COVID-19 pandemic and related restrictions affected both the provision and quality of neonatal care [9]. There is a paucity of scientific evidence on how to best respond to emergency situations in general, with the recent pandemic representing an unprecedented challenge for healthcare professionals and parents alike. Through this study we aimed to evaluate the condition of newborns of mothers with SARS-CoV-2 infection and the immunity of both mothers and neonates, as well as the impact on the mother caused by the separation from the newborn at birth, the impact it had on breastfeeding, and development during the first months after birth.

## 2. Materials and Methods

### 2.1. Study Design and Population

A longitudinal, prospective study was conducted to evaluate newborns of mothers with SARS-CoV-2 infection confirmed before delivery by RT-PCR test, and admitted between 1 April 2020 and 31 March 2021 in the Neonatology Department of the 1st Obstetrics Clinic, Cluj Napoca, Romania. The study group included 89 mothers and their 90 neonates whose gestational age was between 32 and 42 weeks, and who originated from one twin pregnancy, with the rest singleton pregnancies.

### 2.2. Evaluation Parameters

#### 2.2.1. Maternal Cohort

Data were collected on the symptoms of the mothers on admission, together with RT-PCR tests from nasal swab samples. All mothers reported in the study had a positive RT-PCR test at the time of delivery.

Two months after giving birth, the mothers enrolled in the study were asked to complete an online questionnaire or, if unable to do so online, a telephone questionnaire. In this questionnaire, they were asked questions related to the testing of maternal immunity after the SARS-CoV-2 infection and whether they had developed antibodies. Subsequent questions related to breastfeeding the newborn: when it was initiated, the maintaining or interruption of breastfeeding, and the effect of hospitalization on their physical and mental condition. There were also questions related to the outcome of the infant after discharge in terms of whether s/he had contracted the SARS-CoV-2 infection in the first months after discharge from the maternity ward.

During admission after delivery, all mothers were isolated and did not come into contact with their newborns during the entire hospitalization period and thus did not have the opportunity to breastfeed. The length of hospitalization and isolation were dependent on the mother’s symptoms, but also on the existing quarantine rules at the time of birth, which varied depending on the progression of the pandemic.

#### 2.2.2. Evaluation of Neonates

The study group consisted of 90 neonates born to mothers with a positive RT-PCR test, admitted in our unit between 1 April 2020 and 31 March 2021. The gestational age of neonates was between 32 and 42 weeks, and all had RT-PCR testing for SARS-CoV-2 on samples collected at birth from nasal swabs. The newborns were isolated in a special area until receiving the result of the PCR test, after which they were transferred to the intermediate room, where they were cared for until discharge from our service. Their clinical development was monitored and laboratory determinations performed on first day of life, as follows: blood count; inflammatory marker: C-reactive protein (CRP); liver function: aspartate aminotransferase (AST), alanine transaminase (ALT), lactic dehydrogenase (LDH) and creatinine. At two months of age, the children’s type of feeding was analyzed, and the presence of IgG anti-SARS-CoV-2 antibodies was determined. The duration of the hospitalization and the moment of the mother’s contact with the newborn with regard to its influence on breastfeeding, were also analyzed. The data were obtained from the questionnaires completed by the mothers two months after giving birth.

##### RT-PCR Testing

Testing was performed both on the mother and the newborns with the reverse transcription-quantitative PCR (RT-qPCR) technique from the upper airway specimens (nasopharyngeal swabs). Testing of the newborns was conducted on their first day of life. Sampling was performed by a physician wearing appropriate personal protection equipment (PPE) in the quarantine room. Each nasopharyngeal swab was collected in the Nasopharyngeal Sample Collection Kit for Viruses (Dakewe Bio-engineering Co., Ltd. Boston Industries, Boston, USA) and transferred to the laboratory. Nucleic acid isolation was performed with a STARlet IVD Microlab (Hamilton) automatic extractor. A cartridge was placed in a Bio-Rad CFX96 (Bio-Rad Laboratories, Inc.) for polymerase chain reaction, via RT-qPCR. The result was scored as ‘positive’ or ‘negative’.

#### 2.2.3. Follow-Up Questionnaire

The questionnaire consisted of questions (with prespecified answers, single or multiple response answers option) about the time of onset of maternal symptoms related to the time of birth: (1) description of maternal symptoms; (2) data on breastfeeding (age of the child at the time of contact with the mother, time of breastfeeding initiation, previous experience in breastfeeding, maintaining breastfeeding at the time of completing the questionnaire); (3) maternal and infant immunity—evaluation of maternal immunoglobulins; (4) mental health; and (5) experience during hospital stay. The questionnaire was completed online or by phone. The mothers were informed that their newborn would be evaluated at two months after birth, to determine the infant’s antibody titer (Figure 1).

### 2.3. Statistical Analysis

For statistical analysis, the SPSS software package version 25.0 (SPSS Inc., Chicago, IL, USA) was used. The acceptable error threshold was *p* = 0.05. The normality of the quantitative data was verified with the Kolmogorov–Smirnov test. In order to describe the continuous quantitative data with normal distribution, the arithmetic mean and the standard deviation (SD) were used, or median, first quartile (q1) and third quartile (q3) for data with distribution. Qualitative data were described using absolute and relative frequencies. Student’s *t* test for independent samples was used to verify the significant difference between means for quantitative data with normal distribution and a non-parametric Mann–Whitney U test, where necessary. A chi-square test was used to check the association between qualitative data and to quantify the effective size of association, a 95% confidence interval (CI) for OR was calculated.

### 2.4. Ethical Consideration

The follow-up questionnaire was completed by the mothers who gave their consent to answer the questions. The study was approved by the research committee of County Emergency Hospital, Cluj Napoca (28825/2021).

## 3. Results

The 90 newborns (88 singletons, and 1 pair of twins) enrolled in the study had negative RT-PCR test results at birth. There was no vertical transmission of SARS-CoV-2 infection in the study group. None of the newborns presented clinical symptoms suggestive of SARS-CoV-2 infection after admission to the neonatal ward. In the case of mothers, 55 (61.80%) had no symptoms related to the SARS-CoV-2 infection.

The maternal age of the women enrolled in the study ranged from 22 to 40 years (mean/standard dev 28.91 (5.06)). Symptoms such as anosmia and ageusia were reported in 19 cases of infection, followed by cough (8 cases), asthenia (6 cases) and fever (5 cases). In two cases, pneumonia was present at admission and these cases also presented respiratory failure. To obtain information about breastfeeding and their post-COVID immunity, the mothers enrolled in the study were contacted two months after delivery and asked to complete the survey. Figure 2 shows the make-up of our maternal study group. Of the 89 mothers enrolled in the study, 16 did not respond to the request. In 61 of those who responded, antibodies related to SARS-CoV-2 infection were not determined. From the study group, 12 mothers had their IgG levels determined, with 9 showing IgG antibodies.

The twin pregnancy was delivered by cesarean section (2%); out of 88 singleton pregnancies, 70 (78%) were delivered by cesarean section and 18 (20%) were vaginal deliveries.

The characteristics of newborns enrolled in the group are presented in Table 1.

The laboratory parameters evaluated after birth are shown in Table 2.

We analyzed the influence of maternal symptomatology on the monitored laboratory parameters. We wanted to highlight whether the symptomatic maternal infection had any impact on the laboratory parameters of the newborn. The data are shown in Table 3.

Among the 34 (38% of total maternal cohort) mothers who were symptomatic, 19 (21%) presented anosmia and ageusia, the symptoms with the highest recorded proportion. Severe forms of the disease with pneumonia and respiratory failure at the time of birth were present in 2 women participating in the study.

We found that neonates from mothers with anosmia and ageusia had a higher value of WBC and lymphocytes (*p* = 0.06, marginally significant and *p* = 0.04,) and AST (*p* = 0.08, marginally significant).

As a symptom, cough (8.9%) was more often related to an advanced maternal age (30.04 vs. 26.88, *p* = 0.024) and had no influence on the laboratory parameters of the newborn.

There were no significant differences (*p* = 0.17) in terms of hospital stay for neonates from mothers with symptoms (average of 7.2 days) compared to those who did not have symptoms (average of 6.18 days).

Breastfeeding was started by 57.3% of mothers after discharge and only 41,6% of whole study group continued at the follow-up visit. Extending the duration of hospitalization significantly influenced breastfeeding, thus newborns who were not breastfed had an average duration of hospitalization of 12.5 days compared to those who were breastfed, where the duration of hospitalization was 5.02 days (*p* = 0.01). The moment of initiation of breastfeeding in the case of newborns who were breastfed was not influenced by the mother’s breastfeeding experience. The experience related to the hospital stay was negative in 40 cases (54.8% of respondents). Mothers were emotionally affected by separation from the newborn and inability to breastfeed. Mothers who described a negative experience (physical or emotional) during hospitalization associated a 2.42 times higher risk of not continuing breastfeeding, compared to those who did not describe negative experiences (*p* = 0.05, OR = 2.42 (95%CI 1.2–6.31)) (Figure 3).

We had 16 (15.73%) non-respondents to the questionnaire. From the questionnaire applied to the mothers, it was found that none of the newborns had a SARS-CoV-2 infection in the first months after birth.

From the 12 neonates tested, only 6 had IgG antibodies; in 61 cases (67%) the antibodies were not determined because of maternal refusal. The IgG antibody screening for neonates and mothers was conducted after discharge.

## 4. Discussion

The purpose of the study was to assess the development of newborns from mothers with SARS-CoV-2 infection and its correlation with the maternal symptoms and follow-up after two months from discharge, in order to analyze the development of maternal and neonatal immunity, as well as the progression of breastfeeding after discharge. All mothers in the study group were isolated from their neonates during hospitalization, the first contact with them being carried out at home. The study took place in a single center that was designated as a COVID maternity center to serve pregnant women with SARS-CoV-2 infections in the region. Pregnant women with confirmed or suspected SARS-CoV-2 infection were hospitalized in this unit. The unit’s protocol provided for delivery and care until discharge from the maternity ward in the isolation area, without any contact with the neonate during hospitalization.

Unlike other studies, in our study, symptoms were present only in 38.2% of mothers with the SARS-CoV-2 infection [10,11,12]. Other studies have reported that fever, cough, and dyspnea were among the most common symptoms of COVID-19 at the time of hospital admission [13,14]. In our study, anosmia and ageusia were the most frequent symptoms, being found in 19 of the pregnant women with infection. Pneumonia with respiratory failure was present in only 2 mothers in the study, both presenting premature birth, one of whom was pregnant with twins. A systematic review of 16 studies highlighted a variability between 43.5 and 92.0% of asymptomatic pregnant women [15]. The intensive care hospitalization rate of pregnant women with SARS-CoV-2 infection was relatively low at 2.2% in our study, similar to other studies. Thus, Al Matary et al. reported a 3.8% incidence of pregnant women who required intensive care, all being at a gestational age between 28 and 36 weeks [16]. The two cases in our study that presented respiratory failure caused by SARS-CoV-2 infection gave birth prematurely, by cesarean section at 32 and 34 weeks, respectively, due to the respiratory disease. None of the pregnant women enrolled in the study died. The studies demonstrate that the rate of mortality and hospitalizations in intensive care in pregnant COVID-19 women is similar to that in the general population [17,18].

The majority of SARS-CoV-2 positive pregnant women gave birth by cesarean section (78.65%), which shortened the contact period of the staff with the infected patient. The systematic analysis carried out by Capobianco et al. highlighted that the prevalence of cesarean sections ranged from 66.7% to 100% of pregnant women where the pooled percentage of cesarean section was 88% [10]. Several other studies have reported an incidence of over 90% of cesarean births in pregnant women with SARS-CoV-2 infection [19,20,21].

The gestational age of the neonates in the study varied between 32 and 42 weeks. More than two-thirds were born at term, with only 12 (13.33%) premature newborns. Other studies described a higher incidence rate of premature births, but these studies enrolled pregnant women with symptoms present in over two-thirds of cases [11,12,13]. A prospective cohort study carried out by Knight et al. in the United Kingdom found out that about 66% of newborns were preterm infants. The median gestational age at delivery was 38 weeks [17]. The pooled rate of preterm babies in the systematic analysis conducted by Capobianco et al. was 23%, which is higher than our proportion [10]. The predominance of asymptomatic cases in the studied group was the determining factor in the low incidence of premature babies in the group.

Discharge from the maternity ward was carried out usually at over 5 days of age, with relatively high variability between 3 and 21 days, depending on the progression of the pandemic and the restrictions at the time of delivery in terms of the development of maternal symptoms. In other studies, the hospitalization of SARS-CoV-2 positive pregnant women had a median length of 3 days [16,22]. None of the newborns in the study group had SARS-CoV-2 infection; all had negative RT-PCR test results. The newborns whose mothers showed symptoms at the time of birth had a higher number of lymphocytes (*p* = 0.07, marginally significant), without other changes in the laboratory parameters. Other studies have described thrombocytopenia in neonates born from mothers with SARS-CoV-2 infection [23,24,25].

The WHO recommended initiating breastfeeding in mothers with SARS-CoV-2 infection as soon as possible, respecting the infection prevention measures. At the same time, the CDC recommended that: “temporary separation of the newborn from a mother with confirmed or suspected COVID-19 should be strongly considered to reduce the risk of transmission to the neonate”. As there were multiple contradictions in recommendations on how to deal with SARS-CoV-2 positive mothers, as well as periods with a large variation in the number of infection cases, the isolation and quarantine measures employed in our unit varied during the study. Breastfeeding was not initiated for any newborn whose mother was infected with SARS-CoV-2 until after their discharge. In the study conducted by Kostenzer et al. on the COVID-19 Zero Separation Collaborative Group, more than 20% of respondents indicated that absolutely no family member (including the parents) or carer were allowed to be present with their newborn [26]. Of the 1800 participants in this multicenter study, 15% were not allowed to visit the newborn at all during their hospitalization. The presence of parents and interaction with the newborns depended on the severity of local policy restrictions. In the case of the neonates enrolled in our study, parent access was not allowed during their hospitalization. Most of the fathers and siblings were quarantined at home, having been in contact with the infected mother or having a positive test, which required their quarantine according to the restrictions in force. The absence of contact between the mother and the newborn implicitly denoted that the newborns in the study group could not hear the mother’s voice, or feel or smell her, potentially influencing bonding and attachment, as well as the newborn’s neurological development [27,28].

Through the questionnaire carried out two months after discharge, we sought to analyze the success of breastfeeding in mothers with SARS-CoV-2 infection, and the impact of the separation of the mother from the newborn on breastfeeding. Over half of the mothers breastfed the newborn after hospital discharge (57.3%). After breastfeeding was initiated, the trend decreased, so that after two months the number of those who continued breastfeeding decreased. Breastfeeding experience from previous pregnancies had no impact on breastfeeding in the current study (*p* > 0.05).

What the study highlighted by surveying the mothers was the negative impact of hospitalization on the maternal emotional status, caused by the separation of the mother from her newborn. Although only 43.95% reported negative emotional and physical experiences during the hospitalization, we must bear in mind that 16 mothers (15.73%) refused to answer the questionnaire. The negative emotions experienced by them could have influenced their attitude toward the request to complete the questionnaire. These negative emotions influenced the attitude toward breastfeeding. Thus, in mothers who described negative emotions related to hospitalization, breastfeeding was significantly more reduced compared to those who were not emotionally affected during the hospitalization period (*p* = 0.05, OR = 2.42). We must consider that the duration of hospitalization was significantly shorter (4.77 days) in patients who did not describe negative emotions than in those who described emotional/physical impairment (7.88 days) (*p* = 0.01).

In other studies, the questionnaires answered by the enrolled parents highlighted a lack of adequate mental support and a lack of adequate information in more than 50% of the participants. Participants who were exposed to quarantine measures had a higher incidence of negative experiences [26,29].

A study carried out by Rodrigues et al. highlighted that the lack of adequate information about breastfeeding during hospitalization increased the rate of breastfeeding abandonment. In the same study, some of the surveyed mothers perceived the confinement measures as a factor that increased the time allocated to the child and implicitly breastfeeding [30]. However, most mothers reported that the lack of adequate information, lack of support, and the reduction in the number of appointments with medical professionals exerted a negative influence on their breastfeeding practice [24,30,31,32]. The policy of separating newborns from their parents, resulting in a lack of contact with the newborn, was a potentially harmful measure which was implemented with little supporting evidence. Pregnant women should benefit from measures for pregnancy, birth, and postpartum-related care even during crisis or emergency situations such as a pandemic and the inherent lockdown, in accordance with human rights [23]. The World Health Organization and UK guidance therefore advocate to keep newborns with their mothers because of its importance for breastfeeding, bonding and development of the infant. The separation of newborns from their parents has been proven to impair developmental outcomes [25]. In line with WHO’s Baby Friendly Hospital Initiative [33] and the UNICEF Mother Friendly Hospital Initiative [34] on breastfeeding, rooming-in for 24 h a day and skin-to-skin care after delivery is beneficial to the health and development of neonates [33] by facilitating bonding, which should not be disregarded, even during critical periods.

Studies have shown the heterogeneity of antibody responses to SARS-CoV-2. Published reports show long-lasting IgG and IgG1 antibody responses which correlated with disease severity for at least six months post infection [9].

In our research, we aimed to evaluate passive immunity by measuring IgG in the newborn and checking for the presence of IgG in the mother. The information obtained on the post-infection IgG immunity in the mother and the newborn was relatively little, compared to the size of the cohort. Only 12 newborns were tested for the level of IgG at follow-up. Of these, only six presented IgG. In the remainder of the patients, the mothers refused antibody titer testing. Out of the 12 mothers tested, 9 presented IgG after a SARS-CoV-2 infection episode.

One of the limitations of our study is the method used to survey the mothers, which was either online or by telephone. Furthermore, during the study, the degree of restrictions varied and were not uniform. This could generate bias at the time of the start of breastfeeding and in the emotional perception of separation from the newborn. We therefore cannot consider a linear gradient indicating that the more restrictive the measure, the more severe the implication for neonatal care.

The study allowed the analysis of the impact on breastfeeding after the complete separation of the neonate from the mother after birth, and the emotional impact on the mother determined by these conditions of hospitalization. Not a single newborn was infected after discharge from the maternity hospital, even if the mother still had symptoms or a positive test on discharge from the maternity hospital.

The role of parents in the care of a newborn is crucial, the separation of newborns from their parents is not justified by the risk of infection. Even more so, it can even be harmful. Parents and their children must stay together, the implementation of zero separation measures being essential to eliminate the suffering of the youngest and most vulnerable population in society.

The long-term evaluation of the neuropsychic development of children who were separated from their mother at birth will bring to light relevant information on the impact of this separation and allow the rethinking of restrictive measures in pandemics or crisis situations in the future.

## 5. Conclusions

Maternal SARS-CoV-2 infection had an impact on newborns. Although there were no reported cases of vertical transmission, nor multiple cases of severe infections in newborns, the experience of maternal hospitalization under restrictive conditions due to the pandemic and the separation of the newborn from its mother had a negative impact on the maternal psycho-emotional balance and on breastfeeding. Future multicenter studies will be able to highlight in an appropriate way the impact of isolation measures on breastfeeding, as well as the long-term effects on children exposed to these measures during the perinatal period. The long-term evaluation of the neuropsychic development of children who were separated from their mother at birth will result in relevant information on the impact of this separation and allow the rethinking of restrictive measures in pandemics or crisis situations in the future.

## Figures and Tables

**Figure 1 children-10-00771-f001:**
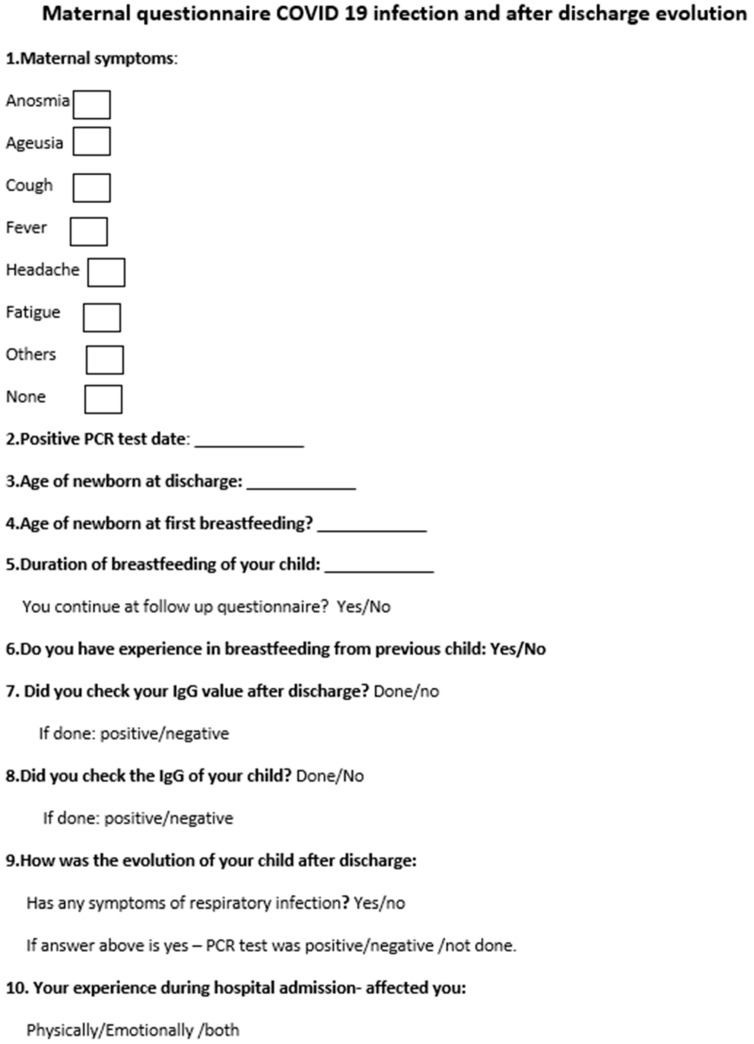
Follow-up questionnaire.

**Figure 2 children-10-00771-f002:**
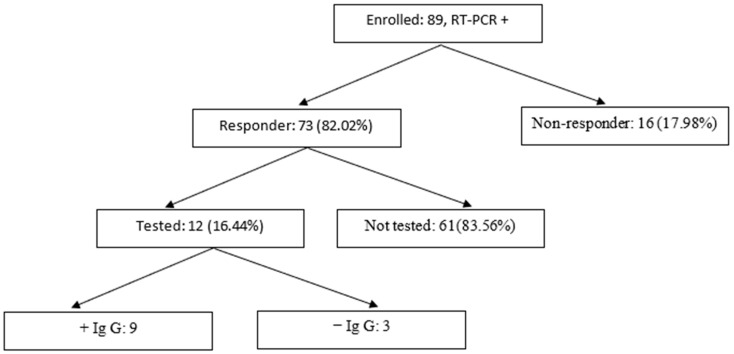
The maternal cohort.

**Figure 3 children-10-00771-f003:**
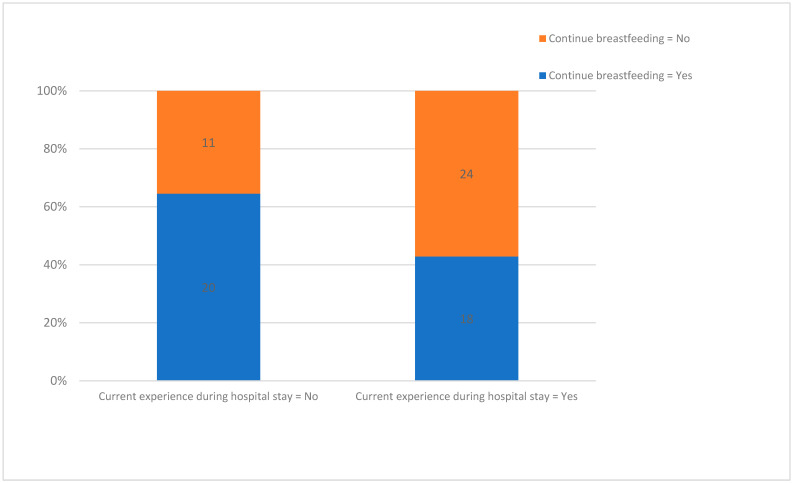
Experience of hospital stay on breastfeeding.

**Table 1 children-10-00771-t001:** Characteristics of newborns in the study group.

Variable	Minimum	Maximum	Mean (std.dev) or Median (q1–q3)
Gestational age (weeks)	32	42	39 (38–39)
Birth weight (g)	1620	4380	3172.64 (553.66)
Apgar 1′	5	10	10 (9–10)
Apgar 5′	7	10	10 (9–10)

**Table 2 children-10-00771-t002:** Laboratory parameters of the newborn study group.

Variable	Minimum	Maximum	Mean (std.dev)
Hb (10 × 12/L)	12.5	20	16.44 (1.82)
Ht (%)	36.2	58.8	47.53 (5.47)
WBC (10 × 19/L)	4.54	31.57	17.63 (6.70)
Ne %	23.9	81.5	59.92 (13.61)
Ne (10 × 19/L)	2.63	24.13	11,060.59 (5790.91)
Ly %	11.2	67.2	29.76 (11.88)
Ly (10 × 19/L)	2.89	10.57	4663.91 (1959.21)
Mo %	1.6	15.5	7.44 (3.09)
Mo (10 × 19/L)	1.90	2.21	1.24 (0.52)
Plt (10 × 19/L)	241	439	272.41 (90.388)
CRP (mg/dL)	0.01	0.86	0.25 (0.25)
AST (U/L)	20	253	68.12 (43.62)
ALT (U/L)	4	347	26.7 (57.96)
Creatinine (mg/dL)	0.14	1.14	0.76 (0.24)
LDH (U/L)	281	2595	572.06 (375.21)

Hb = hemoglobin; Ht = hematocrit; WBC = white blood cells; Ne = neutrophiles; Ly = lymphocytes; Mo = monocytes; Plt = platelets; CRP = C-reactive protein; AST = aspartate aminotransferase; ALT = alanine transaminase; LDH = lactic dehydrogenase.

**Table 3 children-10-00771-t003:** Correlation of newborns’ laboratory parameters with maternal SARS-CoV-2 infection related symptoms.

Parameters	With Symptoms Mean Values (st dev)	W/o Symptoms Mean Values (st dev)	*p*-Value
Ht (10 × 12/L)	6.01 (1.73)	5.23 (1.11)	0.489
WBC (10 × 19/L)	6.92 (1.99)	6.47 (1.38)	0.212
Ne (10 × 19/L)	6.12 (1.76)	5.58 (1.19)	0.284
Ly (10 × 19/L)	1.90 (0.549.)	1.87 (0.40)	0.07
AST (U/L)	26.44 (7.63)	48.9 (10.67)	0.052
ALT (U/L)	15.5 (8.26)	33.1 (72.24)	0.41
CRP (mg/dL)	0.15 (0.14)	0.3 (0.27)	0.048
Creatinine (mg/dL)	0.27 (0.08)	0.22 (0.05)	0.915

Ht = hematocrit; WBC = white blood cells; Ne = neutrophiles; Ly = lymphocytes; AST = aspartate aminotransferase; ALT = alanine transaminase; CRP = C-reactive protein.

## Data Availability

All data generated or analyzed during this study are included in this published article.

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
