# Peer review of "The Effect of Maternal SARS-CoV-2 Infection on Neonatal Outcome"

_children, 2023, doi:10.3390/children10050771_

Round 1
Reviewer 1 Report
Dear Authors, Congratulate for considering such interesting subject. I am truly impressed by taking such subjects as: immunization, the influence of separation into mothers. It would be great if You do also the assessment of psychomotor development of child. I would like You ask to explain and add the references to the sentence first sentence: “Due to the immunological peculiarities associated with pregnancy, pregnant women represent a category with increased risk for the SARS-CoV-2 infection” – line 31 -32. I am impressed that no child was hypothrophic. Could You explain why authors write in results caesarean sec. in capital letter, and in discussion - line 215 in small letter. I suggest to make the rule equal. Could You add why so small amount of people were examined referring to IgG? Could You name the questionnaire You have used to examine parents? Was it the tool You created? Could You translate it into English and add it as an attachment of Figure? I would like to say that discussion is very mature and You explain all the presented results. Congratulation!
Reviewer 2 Report
Dear Authors,
The study could be interesting, a large group of mothers and children were enrolled in the study, but it contains a lot of errors and inaccuracies and omits clinical aspects of children.
1. Please standardize when a mother's history was collected about her child's health status and feeding method
Line 26- first 3 months
Line 88- at 2 months
Line113- between 2 and 3 months
Line 137- 2 months
Line 189- 2 months
2. Line 69-70: “There were also questions related to the evolution of the infant after discharge, respectively if s/he contracted the SARS-CoV-2 infection in the first months after discharge from the maternity ward”- these issues were not mentioned in the study results.
3. Table 1. It is worth adding data on the number of premature babies, and the number of babies born with asphyxia. Is the weight distribution parametric? Shouldn't there be a median instead of a mean?
4. There is no information on which day of life the blood was drawn for neonatal screening.
5. No information on the length of hospitalization and the course of adaptation in general and according to the presence or absence of maternal symptoms.
6. No information on the clinical problems of children after discharge home.
7. Line 167- p=0.06 and 0.08 are not statistically significant
8. When was IgG testing done in children and mothers- still during hospitalization? After discharge?
9. Line 239- wrong conclusion, p is not statistically significant.
10. Line 314- 9, not 8 mothers had IgG.
Reviewer 3 Report
I would like to congratulate the authors on the selection of this novel and the interesting aspect of cardiology. The authors have provided good evidence to support their conclusion with well constructed and meticulously written manuscript.
However, I would like to bring attention to the following points
1. Please elaborate statistical section on the methodology
2. In the results section provided results in graphical format can be helpful for readers
3. In the limitations section more succinct presentation is needed.
Round 2
Reviewer 2 Report
Dear Authors,
Thank you for considering the reviewers' comments. The study is clearer. However, I have a few more comments:
1. Among several objectives, the aim of the study was to assess the impact of maternal and neonatal isolation on the development of children in the first months of life (line 51), while there is no information on this topic in the entire study. Has this issue been studied? If not then please remove this objective from the description.
2. Table 1 (Characteristics of newborns) contains information on the age of mothers. Since this information regarding mothers is also included in the text, it should be removed from the table.
3. The units in which the blood morphology parameters are presented (Tables 2 and 3) are incorrect, e.g. WBC 17622.94 x 1019/L instead of 109/L, Hb 16.44 x 1012/L instead of 102.
4. Please provide information on what was the average time of hospitalization of newborns: Lines 190-191 vs 195-196 (the average time cannot be 12.5 days since the longest stay was 10 days).
5. Line 112 “…maternal immunity…” should be maternal and infant immunity.
6. Figure 3: should be “continue” instead “continuu”.